# Multiway Analysis of the Electrochemical Oxidation Pathway of a Lignin Using Chemometrics

**DOI:** 10.3390/molecules30214305

**Published:** 2025-11-05

**Authors:** Gobind Sah, John A. Staser, Peter B. Harrington

**Affiliations:** 1Department of Chemistry & Biochemistry, Ohio University, Athens, OH 45701, USA; gs549819@ohio.edu; 2Department of Chemical and Biomolecular Engineering, Ohio University, Athens, OH 45701, USA; staser@ohio.edu

**Keywords:** lignin valorization, bimetallic oxidative catalysis, multiway GC-MS analysis, factor analysis, product identification

## Abstract

The electrochemical oxidation mechanism of biopolymer lignin is challenging to characterize due to its complex structure. Controlling the oxidation process is crucial for ensuring the economic feasibility of electrochemical depolymerization of lignin, as it often generates numerous undesirable compounds. Regulating the depolymerization process can lead to the production of high-yield aromatic compounds, such as phenols and carboxylic acids. In addition to the depolymerization of lignin by the electrocatalyst, hydroxyl radicals (^•^OH) during the electrochemical oxidation could also depolymerize lignin. Previous studies have reported that ^•^OH forms during electrochemical oxidation; however, it is still uncertain whether these radicals or electrocatalysts are responsible for depolymerizing lignin. This study investigates the pivotal issue of whether the depolymerization process is driven by ^•^OH or by a direct electrochemical route. In this study, lignin compounds were electrochemically oxidized using a nickel-cobalt (Ni-Co) electrocatalyst at several electrode potentials, and the oxidized products were analyzed using headspace solid-phase micro-extraction gas chromatography–mass spectrometry (SPME-GC-MS) and factor analysis (FA). Electrochemical depolymerization of lignin yielded mainly phenolic compounds (e.g., tert-butyl phenols), phthalate esters (e.g., dibutyl phthalate, bis(2-methylpropyl) phthalate), furan derivatives (e.g., 2-butyltetrahydrofuran), and short-chain carboxylic acid esters. This work has successfully predicted that both electrocatalyst and ^•^OH radicals contribute to the electrochemical depolymerization of lignin. Radical-mediated depolymerization yielded a broader range of products.

## 1. Introduction

According to the European Commission’s “World Energy, technology, and Climate Policy Outlook 2030 (WETO)”, global energy consumption is projected to increase from 1.21 to 1.71 gigatons of oil equivalent between 2010 and 2030, while carbon dioxide emissions are expected to rise from 29.3 gigatons to 44.5 gigatons during the same timeframe [1,2]. Currently, most chemicals are generated from non-renewable fossil fuels such as petroleum and coal, resulting in the greenhouse effect, climate change, and environmental damage. One of the most effective ways to reduce the use of fossil fuels is to produce chemicals and materials from renewable resources using sustainable biomass like lignin [3,4,5]. Utilizing lignocellulose, the most abundant renewable feedstock on Earth, is essential for the development of sustainable biorefineries capable of producing valuable bioproducts and energy [6]. Governments across the world are encouraging the adoption of bio-based alternatives to replace fossil fuels.

By 2030, the U.S. Department of Agriculture and Energy aims to have 25% of commodity chemicals and materials made from bio-based materials, up from 5% in 2005 [7,8]. About 30% of the carbon resources on Earth come from lignin, the second most abundant biomaterial [9,10]. Because lignin is rich in phenylpropanoid content and has significant annual production from the pulp and paper industry, it has been considered a feedstock for manufacturing petroleum-based chemicals such as phenolic compounds and aromatic hydrocarbons [10,11]. Lignin has also been utilized as a bioadsorbent to remove toxic dyes from contaminated water [12]. However, due to lignin’s complex structure and the lack of efficient depolymerization methods, it is either discarded or used as a substandard fuel in biorefineries and pulping mills [13]. A low-cost depolymerization method could transform lignin waste into valuable, high-yield chemical compounds like aromatic phenols. In addition to value-added chemicals, lignin has been valorized into various products, including carbon fibers, resins, plastics, adsorbents, and energy storage devices [14,15].

Several depolymerization techniques have been developed to scission lignin into lower-molecular-weight aromatics (LMWAs) [10], including catalytic hydrodeoxygenation [16,17], pyrolysis [18], microwave-assisted depolymerization [19,20], gasification in supercritical water [21,22], oxidative lignin depolymerization [23,24], hydrocracking [25,26], and hydrothermal fragmentation and condensation [27].

All depolymerization methods follow one of the three available routes: catalytic oxidation, catalytic reduction, and hydrolysis [10,28,29]. The main issue with reduction is the degradation of aromatic functionality in the lignin conversion products, whereas poor rates of aromatic compound formation plague hydrolysis [11]. Although catalytic oxidation techniques may yield a significant yield of functionalized aromatic compounds, most catalytic oxidation reactions require complex procedures and toxic additives, presumably adding to the expense of the system [10,28].

Hence, there is a need for a low-cost, nonpolluting depolymerization process. Electrochemical approaches for lignin conversion are highly promising as they are scalable, free of external oxidants or reductants, and can be conducted under mild conditions [30]. Bailey and Brooks initially attempted to electrochemically oxidize lignin in the mid-1940s [31,32]. Several studies have reported the electrochemical oxidation of lignin [33,34,35,36,37,38,39,40,41,42,43]. Compared to other depolymerization processes, electrochemical oxidation [10,34,44,45,46,47] is cost-effective [46] and environmentally friendly [38,48]. The participating electrons in electrochemical oxidation do not function as pollutants or produce non-utilizable byproducts [47,49]. An added benefit of electrochemical oxidation is the production of hydrogen at the cathode [50,51]. Additionally, lignin depolymerization using electrochemical oxidation could be a viable strategy for reaching carbon neutrality [30].

While electrochemical oxidation is promising for lignin depolymerization, electrode fouling and high cost hinder its use. Thus, the development of cost-effective electrocatalysts for lignin oxidation is crucial. This group first studied the effect of three electrocatalysts (nickel (Ni), cobalt (Co), and Ni-Co bimetallic electrocatalysts) on lignin oxidation [34]. We later developed an inexpensive Co core/Pt partial shell alloy [33] and Ni-Co electrocatalysts [52] to oxidize lignin.

The analytical methods for lignin analysis are diverse. Chemometric methods such as principal component analysis (PCA), principal component analysis-quadratic discriminant analysis (PCA-QDA), and the generalized standard addition method (GSAM) have been used for lignin analysis [53,54,55,56]. For instance, Prothmann et al. [53] used PCA-QDA with ultra-high performance liquid chromatography/high-resolution multiple-stage tandem mass spectrometry (UHPLC-HRMS^n^) to identify lignin oligomers in kraft lignin. Our group used GSAM to quantify the electrolytic depolymerization of lignin using an ultraviolet plate reader [56]. Similarly, Benar and Gonçalves [54] utilized PCA in conjunction with Fourier transform infrared (FTIR) spectroscopy to analyze lignin products obtained during the hydroxymethylation of sugarcane lignin. Likewise, Fink et al. [55] used PCA with attenuated total reflectance Fourier transform infrared (ATR-FTIR) spectroscopy for lignin analysis.

Wang et al. investigated the electrocatalytic degradation of aspen lignin by Pb/PbO_2_ electrodes [35]. They claimed that ^•^OH radicals depolymerized lignin. They considered previous research conducted by Quiroz et al. [57] on the oxidation of *p*-nitrophenol with similar electrodes.

Cabrera et al. [33] oxidized lignin using a Co core/Pt nanoparticle electrocatalyst and claimed ^•^OH radicals mediated the scission of lignin. Zhu et al. used a Ni-Co/C bio-metallic catalyst for lignin depolymerization and analyzed the products using gas chromatography–mass spectrometry (GC-MS) [37]. The ^•^OH radical has an essential role in various anodic reactions [58,59]. The ^•^OH radicals generated at the anode oxidize organic compounds [31].

Numerous studies [33,35,36,57] have suggested the potential formation of ^•^OH radicals that depolymerize lignin through radical attack. However, the studies have not proved whether lignin depolymerizes due to the generated ^•^OH radicals or through direct electrochemical oxidation at the electrocatalyst’s surface. Our previous work on benzyl phenyl ether revealed ^•^OH radical attack and direct electrochemical pathways contributed to the decomposition [60].

Our present work explicitly addresses the characterization of the electrochemical oxidation pathway for lignin. We found similar behavior for lignin, as the electrocatalyst and ^•^OH radicals are both involved in the depolymerization. Beyond elucidating the electrochemical oxidation process of lignin, our work putatively identifies the products with mass spectral library searches.

## 2. Results

The lignin sample underwent electrochemical oxidation at three distinct electrode potentials (0.6 V, 0.8 V, and 1.0 V vs. Hg/HgO), both in the absence and presence of DMSO (dimethyl sulfoxide). The solution was bifurcated after dissolving the lignin sample in a 1:3 ratio of acetonitrile (CH_3_CN) to 1.0 M sodium hydroxide (NaOH): one was sealed and referred to as the neat sample. The other samples were subjected to electrochemical oxidation and referred to as the oxidized samples. The samples destined for oxidation were split further into two, with one portion supplemented with DMSO.

The results have been categorized into three subsections based on the three oxidation potentials. Each section includes figures that compare the total ion current between oxidized without DMSO and oxidized with DMSO, along with factor analysis scores, retention time loadings, and mass spectral loadings. Score plots are presented using factor components (FC) are rotated so that the treatment effect aligns with the first factor. The purpose of the rotation is to facilitate the interpretation of the factor’s variable loadings, which indicate the importance of each peak in the chromatographs or mass spectra. Each point in the score plot corresponds to a sample, with separation along the factor indicating differences in composition caused by the addition of DMSO, i.e., the treatment.

Marginal factor loadings reduce the two-way data, i.e., GC×MS, by summing along the retention time or mass-to-charge ratio axes. Positive and negative loadings relate to the scores, with positive scores having larger positive loadings and negative scores having negative loadings. The magnitude of the loading indicates its relative concentration. Together, score and marginal loading plots allow a clear visualization of complex data.

### 2.1. Study of the Effect of Oxidation Potential 1.0 V vs. Hg/HgO

The oxidation potential of 1.0 V vs. Hg/HgO is the highest among the three tested oxidation potentials. Therefore, more oxidized compounds were expected compared to other potentials. Figure 1 has the total ion currents (TICs) of the five replicates of $O_{1.0}$ oxidized (without DMSO) sample overlaid on the TICs of the five replicates of $O_{1.0}^{D}$ oxidized (with DMSO) sample. Samples with and without DMSO had different peaks. In cases with overlapping peaks, variations in peak intensity can be observed. The peaks at retention times 5.01, 5.92, 6.41, 7.44, 7.71, and 8.86 min are due to interferences that may have originated from the solvent, SPME fiber, or GC column.

Figure 2a has the factor analysis scores for both $O_{1.0}$ and $O_{1.0}^{D}$ samples. The confidence intervals of the different samples do not overlap, indicating that there is a significant difference between samples oxidized with and without DMSO. The retention time loadings Figure 2b and mass spectral loadings Figure 2c allow for the interpretation of the score plot. Retention time loadings provide insights into the varying contributions of compounds at different retention times, which in turn affects the overall variation exhibited in the score plot. Loadings near −1 or +1 indicate a significant impact of the variable on the component, whereas loadings close to zero suggest a marginal effect. The positive loadings represent the $O_{1.0}$ (red, without DMSO) oxidized samples. In contrast, the negative loadings represent the characteristic features of the $O_{1.0}^{D}$ (blue, with DMSO).

The mass spectral loadings in Figure 2c indicate that in the presence of DMSO (^•^OH quenched), the mass fragments are primarily formed at a lower *m*/*z* ratio, whereas in the absence of DMSO (^•^OH not quenched), the fragments are formed over a wide range of *m*/*z* ratios. The difference in the fragmentation pattern of oxidized samples with and without DMSO indicates the difference in the electrochemical depolymerization pattern of lignin.

The library search of the observed oxidized products (Table 1 and Table 2) allows for identifying the specific types of products generated, hence facilitating differentiation between the treatments. Most compounds fragmented from lignin with DMSO (Table 2) are different from the compounds formed without DMSO (Table 1), except for those that were observed at retention times of 8.01, 9.01, 9.55, 10.78, and 11.06 min. The compounds at retention times 3.67, 4.26, 7.20, 8.14, 8.97, 9.88, and 11.57 min were exclusively present in $O_{1.0}$ (without DMSO). In contrast, compounds at retention times 7.24, 8.14, 8.38, and 11.54 min were observed only in $O_{1.0}^{D}$. Some oxidized products are simple hydrocarbons, while most others are aromatic compounds. The formation of different compounds indicates that in the presence of DMSO, depolymerization follows a different pathway than without it. Without DMSO, the products were associated with *β-O-4* ether bond cleavage, giving phthalate esters, phenolic derivatives, furans, and cyclic esters. With DMSO, however, distinct oxygenated esters, aldehydes, hydrocarbons, and substituted aromatics were observed. This difference occurs because DMSO quenches the hydroxyl radicals and shifts the oxidation pathways, leading to different product distributions.

Compounds obtained from lignin after electrochemical oxidation had a wide range of structures, which suggests that they break in a complex and varied way during electrochemical oxidation. The formation of esters, such as ethoxyacetic acid ethyl ester and 15-octadecatrienoic acid methyl ester, suggests transformations associated with lignin’s aliphatic and aromatic moieties. The occurrence of propanoic acid derivatives, like 2-methyl-2, 2-dimethyl-1-(2-hydroxy-1-methylethyl)-propanoic acid, propyl ester, and 3-hydroxy-2,4,4-trimethylpentyl ester, indicates the involvement of structural modifications linked to hydroxyl groups in lignin. The presence of cyclic compounds, like cyclohexane carboxylic acid, 3-methylene-2-oxo-methyl ester, and 2-oxetane, implies possible ring-forming reactions during oxidation. Additionally, the appearance of phenol derivatives and furan compounds further reflects the diverse nature of the oxidative transformations, and the mechanism for forming these compounds likely involves intricate reactions, including reaction by the ^•^OH radical and rearrangements, resulting in the observed structural diversity.

From the mass spectral loadings in Figure 2c, the score plot Figure 2a, and the difference in the types of compounds oxidized (observed in Table 1 and Table 2) with and without DMSO, it can be concluded that both the ^•^OH radical and the electrocatalyst depolymerize lignin. When there is DMSO, the lignin depolymerization likely occurs at the electrode’s surface. In contrast, in the absence of DMSO, the free ^•^OH radicals dominate the lignin depolymerization.

### 2.2. Study of the Effect of Oxidation Potential 0.8 V vs. Hg/HgO

This section reports oxidized compounds at 0.8 V vs. Hg/HgO. Table 3 and Table 4 enumerate the oxidation products from $O_{0.8}$ and $O_{0.8}^{D}$.

The samples oxidized at 0.8 V vs. Hg/HgO (Figure 3) exhibited similar TICs to the samples oxidized at 1.0 V vs. Hg/HgO (Figure 1), with the only distinction being smaller peak intensities. This trend conforms with the hypothesis that oxidation decreases at lower potentials. From the scores (Figure 4a), there is a significant difference between the samples with and without DMSO, as was the case for 1.0 V. The mass spectral loadings (Figure 4c) depict a similar pattern observed in the case of oxidation potential 1.0 V vs. Hg/HgO. As before, ions observed for the samples with DMSO were at lower mass-to-charge ratios.

Without DMSO, the ions are distributed over a broader range of mass-to-charge ratios. Both lower- and higher-molecular-weight fragments are observed, indicating the difference in the depolymerization pathways. Comparatively fewer oxidized compounds were observed during electrochemical oxidation at 0.8 V vs. Hg/HgO than those oxidized at 1.0 V.

From the above results, the oxidation potential 0.8 V is sufficient to study the electrochemical mechanism of lignin. The observed results (Figure 4, and Table 3 and Table 4) indicate that depolymerization follows direct electrochemical oxidation and fragmentation by ^•^OH radical attack at 0.8 V.

### 2.3. Study of the Effect of Oxidation Potential 0.6 V vs. Hg/HgO

Among the potentials, 0.6 V vs. Hg/HgO was the lowest; hence, a lower degree of electrochemical oxidation of lignin was expected. From the comparison of TICs (Figure 5) of the two sets of oxidized samples, $O_{0.6}$ (red, without DMSO) and $O_{0.6}^{D}$ (blue, with DMSO), only a few peaks are observed, resulting in a reduced number of oxidized compounds (Table 5 and Table 6) compared to the higher oxidation potentials. The lignin exhibits limited conversion during oxidation at 0.6 V vs. Hg/HgO. Given the understanding that conversion at low potentials might be minimal, different oxidation potentials of diverse ranges (low, medium, and high, i.e., 0.6 V, 0.8 V, and 1.0 V vs. Hg/HgO) were selected to understand the effect of potential on electrochemical conversion of lignin.

Even when the potential is low (0.6 V vs. Hg/HgO), the scores do not overlap (see Figure 6a), and the peaks in the chromatographic and mass spectral loadings indicate that different pathways are involved. See Figure 6b,c. From the chromatographic loadings, the free radical pathway is more efficient at depolymerizing lignin. The mass spectral loadings appear smaller because they are spread across a broader range and normalized.

## 3. Materials and Methods

### 3.1. Materials

Lignin (CAS 8068-05-1) was obtained from Sigma-Aldrich (St. Louis, MO, USA). The material was kraft (alkali) lignin with an average molecular weight of approximately 10,000 Da and a moisture content of up to 5%. Sodium hydroxide (NaOH) was purchased from Fisher Scientific (Waltham, MA, USA). All reagents were used as received without further purification. To investigate the role of radical species in the reaction mechanism, dimethyl sulfoxide (DMSO) was utilized as a hydroxyl radical scavenger at a final concentration of 0.2% *v*/*v*. Our group previously determined this concentration to be effective for quenching radicals in studies involving lignin model compounds [60].

### 3.2. Sample Details

The sample details are in Table 7.

### 3.3. Electrochemical Conversion of the Lignin

Full details on the electrocatalyst synthesis and reactor design can be found elsewhere [34,60,61]. The Ni-Co catalyst ink was sprayed onto a standard gas diffusion layer (GDL) to form a gas diffusion electrode (GDE), which functioned as the working electrode for the oxidation reaction. Platinum foil was employed as the counter electrode, while a mercury/mercury oxide (Hg/HgO) served as the reference for controlling the voltage of the working electrode. A compact batch electrochemical reactor was employed using a small beaker with stirring provided by a magnetic stir bar depolymerized the lignin. The potentiostat was a Bio-Logic Science Instruments (Seyssinet-Pariset, France) SP-150. A constant working electrode potential was maintained for 10 h. Three working electrode potentials were used: 0.6 V, 0.8 V, and 1.0 V vs. Hg/HgO. The electrode potentials were selected based on previous studies showing that the target reaction in the biomass-depolarized electrolyzer occurs at lower potentials than the competing oxygen evolution reaction (OER), which is key to “green hydrogen” production [61,62]. This choice aims to enhance lignin oxidation while minimizing the energy-intensive OER, optimizing depolymerization. Please see our earlier reports for more details on selecting conditions such as pH, electrocatalysts, and temperature [34,60].

Using a random block design, five replicates of each lignin sample were measured over five days to ensure that the GC-MS (Shimadzu Scientific Instruments, Columbia, MD, USA) data acquisition was independent of the measurement sequence and time. The pre-oxidized (neat) and post-oxidized lignin samples were extracted using headspace solid-phase microextraction (SPME) fiber (Sigma-Aldrich, St. Louis, MO, USA) and analyzed using GC-MS. The lignin sample was divided into two parts. Dimethyl sulfoxide (DMSO), a radical scavenger, was added to one-half of the samples, and the results were then compared to the other half without the radical scavenger. Introducing DMSO to half of the solution and keeping the other half unchanged makes it possible to assess whether a distinction exists between samples supplemented with DMSO and those not. The aim is to determine whether a direct electrochemical reaction or a radical-mediated pathway drives electrochemical oxidation.

### 3.4. Theory

Principal component analysis (PCA) is a popular unsupervised chemometric method that can be used to examine the GC-MS data and find covariance in the dataset [63]. The principal components underwent orthogonal rotation to ensure the treatment effect, i.e., the addition of DMSO, would load onto a single component, i.e., factor. [64]. This process is factor analysis (FA)and allows the interpretation of the variable loadings on the first factor. Clustering of the scores reveals similarities among the observations. The 95% confidence ellipses reveal statistical significance.

### 3.5. Instrumental Analysis

A manual solid phase microextraction (SPME) holder with a 75 μm polydimethylsiloxane (PDMS) fiber was used to extract volatile components from the lignin solution. The fiber sampled a 10 mL headspace vial containing 30 μL of the lignin solution at a temperature of 70 °C for 20 min. The fiber was retracted into the needle to protect the adsorbed volatiles. The syringe needle was removed from the vial and inserted through the septum of the GC inlet. Every neat (unoxidized) and oxidized sample underwent the same procedure. Five replicates were obtained throughout the subsequent five days. To ensure the syringe fiber was clean each time, the syringe fiber was sampled until no peaks appeared in the total ion chromatogram.

The neat and oxidized lignin samples were characterized using a single quadrupole gas chromatograph–mass spectrometer (GC-MS-QP2010 SE). The ionization voltage was 70 eV. Temperature programming was used on a Shimadzu Rxi-5ms GC column (30 m × 0.25 mm × 0.25 μm). The column temperature was set at 50 °C for 2 min. The temperature then increased at a constant rate of 20 °C min^−1^ until it reached 250 °C, which was maintained for 3 min, resulting in a total GC run time of 15 min. The injection port temperature was set at 250 °C, while the GC-MS interface was maintained at 270 °C. A low-volume glass liner for headspace in split mode was used with a split ratio of 10:1. The column flow rate was 1.6 mL min^−1^, utilizing ultra-high purity helium as the carrier gas. The mass spectrometer ion source temperature was 250 °C and operated with a range of *m*/*z* 35–350. A home-built computer equipped with an AMD Ryzen Threadripper 3970X 32-core CPU processor operating at 3.69 GHz (Windows 11 Enterprise version 21H2 operating system (Microsoft Corp., Redmond, WA, USA)), 32.0 GB of RAM, and 64-bit operating system was used for all evaluations. MATLAB 2021b (MathWorks, Natick, MA, USA) was used for all evaluations. The singular value decomposition function (SVD) was used to obtain the principal components (PCA). The components were orthogonally rotated so as to maximize the projected difference resolution of the scores to obtain the factors.

The compound’s chemical name and type were ascertained by comparing the query spectrum to the reference spectra from the NIST and Wiley databases. In addition to the similarity scores, the library search included the quantitative reliability metric (QRMf) [65]. The quality of each library search result was evaluated using the QRMf, which statistically evaluates the match’s reliability. When the similarity measure is high and the QRMf is low, the result is considered unreliable; conversely, a low similarity measure with a high QRMf may indicate a reliable result.

The GC-MS data file was exported as a CDF file and then converted into MATLAB mat file format using custom in-house functions. The data underwent baseline correction, and the chromatograms were aligned to the average chromatogram of the dataset. Subsequently, the chromatograms were unfolded and normalized to unit Euclidean length. Each dataset underwent a square root transformation, a standard approach to reduce the dynamic range of the mass spectral peak intensities. The advantage of this transformation is that smaller peaks at higher mass are more informative and will have a greater relative weight compared to the larger peaks at lower mass.

## 4. Conclusions

The findings revealed greater oxidation at the highest oxidation potential (1.0 V vs. Hg/HgO), resulted in more oxidized products than at lower oxidation potentials. Thus, a 1.0 V oxidation potential is sufficient and more effective in depolymerizing lignin than lower oxidation potentials.

If no variations were observed among oxidized lignin samples with and without DMSO, it implies that, at the investigated potentials, the mechanism was not predominantly driven by ^•^OH attack but rather by direct electrochemical oxidation. However, significant differences were observed between cases with and without DMSO for all potentials when depolymerizing lignin. The mass spectra revealed that the oxidized ions were smaller when DMSO was present. In contrast, the fragments were spread across a broader range without DMSO. This result suggests that both ^•^OH radicals and direct electrochemical oxidation participate in the electrochemical depolymerization of lignin.

If ^•^OH radicals were solely responsible for depolymerization, quenching them with DMSO should result in minimal to no oxidation. However, several oxidized products in the presence of DMSO also suggest the involvement of direct electrochemical oxidation. The library search results further supported the dual roles of ^•^OH and direct electrochemical oxidation, which had a comparable number of unique compounds in each case (with and without DMSO). Combining FA with SPME-GC-MS proves to be an effective approach for examining the electrochemical oxidation pathway of biopolymer lignin. Furthermore, processing the full two-way GC×MS information is more efficient than analysis of the GC or MS data individually. The results obtained from the analysis indicate that both the direct electrochemical route and a mechanism mediated by ^•^OH radicals depolymerize lignin during electrochemical oxidation.

## Figures and Tables

**Figure 1 molecules-30-04305-f001:**
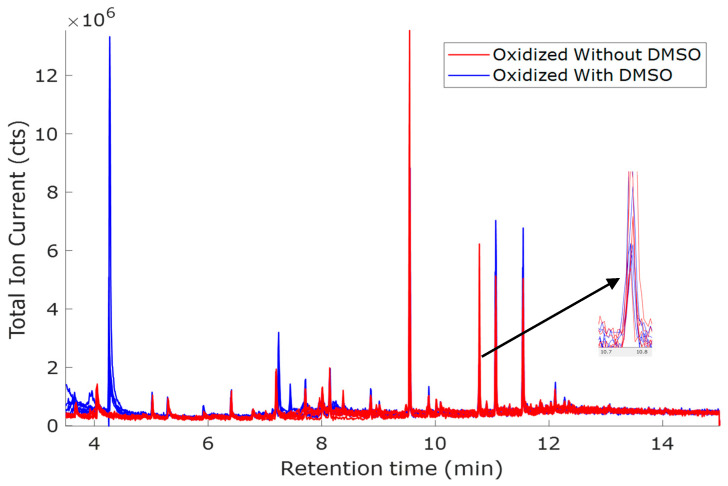
TICs for oxidized $O_{1.0}$ (red, without DMSO) and $O_{1.0}^{D}$ (blue, with DMSO). Solvent: ACN + NaOH (1:3, *v*/*v*); voltage: 1.0 V vs. Hg/HgO; the two oxidized samples were measured with five replicates.

**Figure 2 molecules-30-04305-f002:**
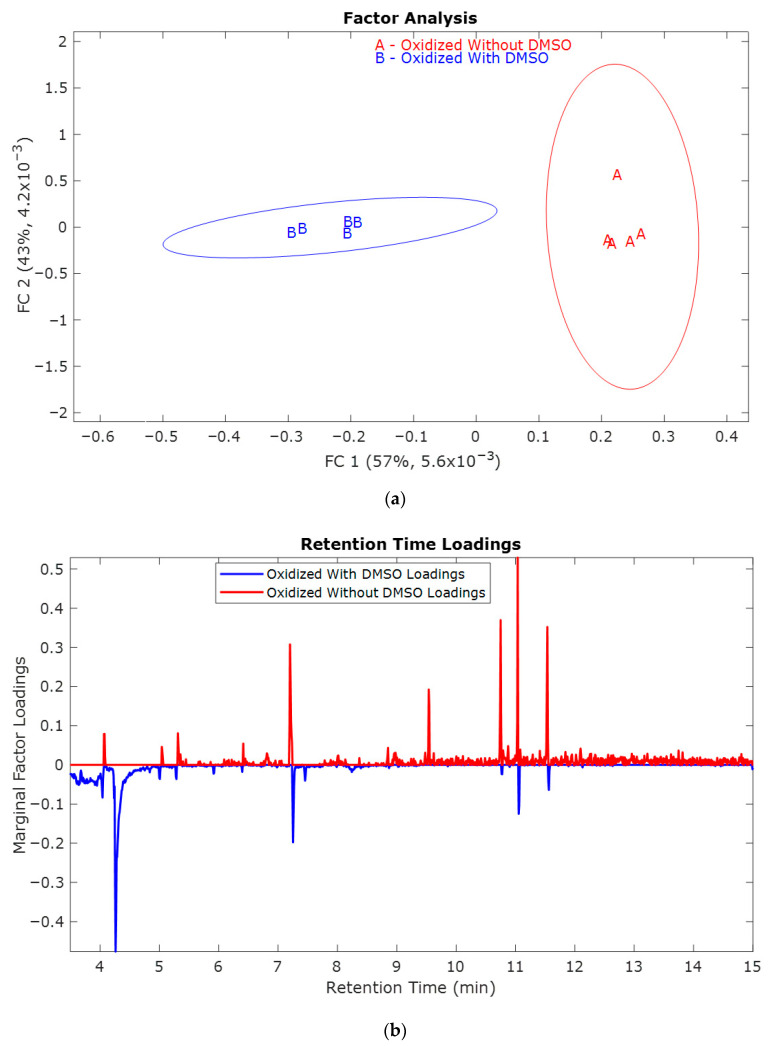
(**a**) Factor analysis to visualize the distributions of the five replicates for $O_{1.0}$ (red, without DMSO) and $O_{1.0}^{D}$ (blue, with DMSO) oxidized samples; (**b**) Retention time loadings and (**c**) Mass spectral loadings of oxidized $O_{1.0}$ (red, without DMSO) and $O_{1.0}^{D}$ (blue, with DMSO).

**Figure 3 molecules-30-04305-f003:**
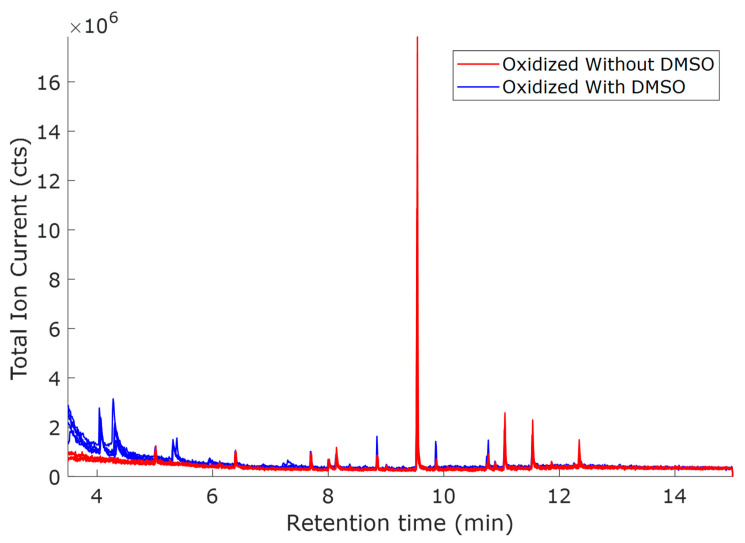
Total ion current (TICs) for $O_{0.8}\;$ oxidized (red, without DMSO) and $O_{0.8}^{D}\;$ samples (blue, with DMSO). Solvent: ACN:NaOH (1:3, *v*/*v*); voltage: 0.8 V vs. Hg/HgO; the two oxidized samples were measured with five replicates.

**Figure 4 molecules-30-04305-f004:**
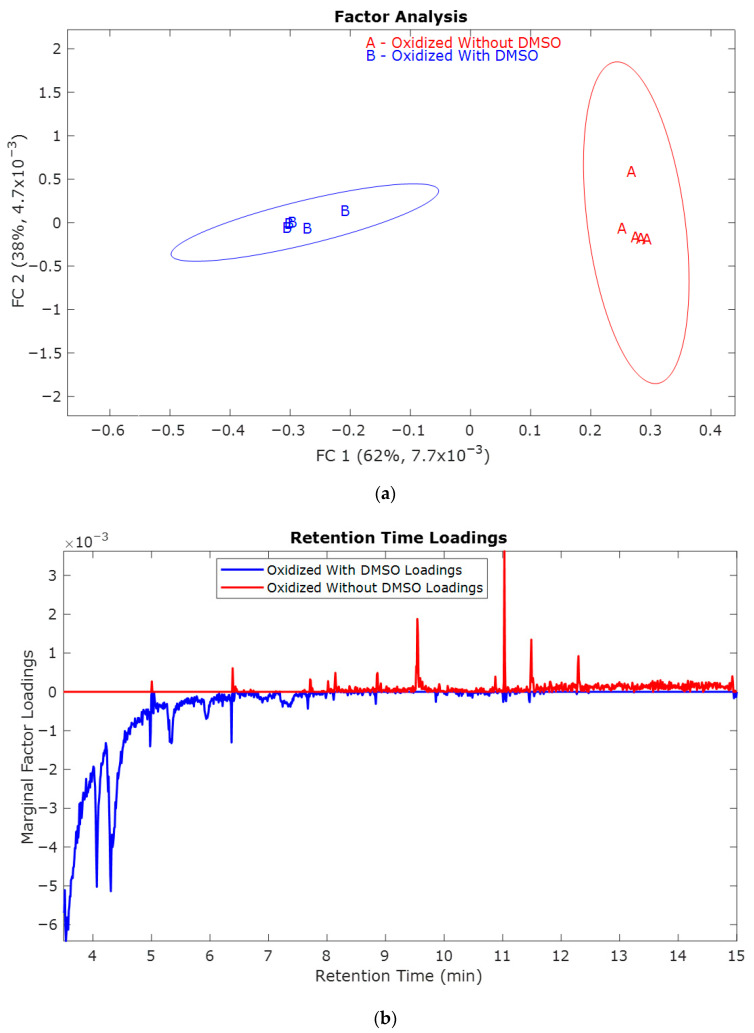
(**a**) Factor of the distributions of the five replicates for $O_{0.8}\;$ (red, no DMSO) and $O_{0.8}^{D}\;$ samples (blue, with DMSO); (**b**) Retention time loadings and (**c**) Mass spectral loadings of $O_{0.8}\;$ (red, without DMSO) and $O_{0.8}^{D}\;$ (blue, with DMSO) samples.

**Figure 5 molecules-30-04305-f005:**
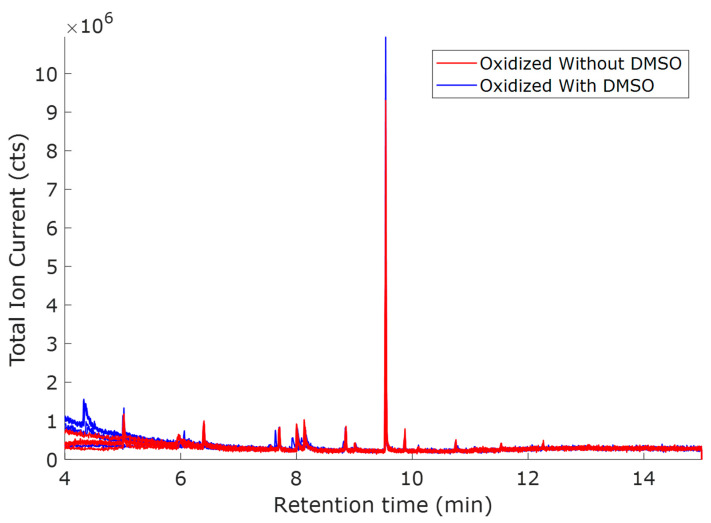
TICs for $O_{0.6}$ (red, without DMSO) and $O_{0.6}^{D}$ blue, with DMSO) oxidized samples. Solvent: ACN:NaOH (1:3, *v*/*v*); voltage: 0.6 V vs. Hg/HgO; the two oxidized samples were measured with five replicates.

**Figure 6 molecules-30-04305-f006:**
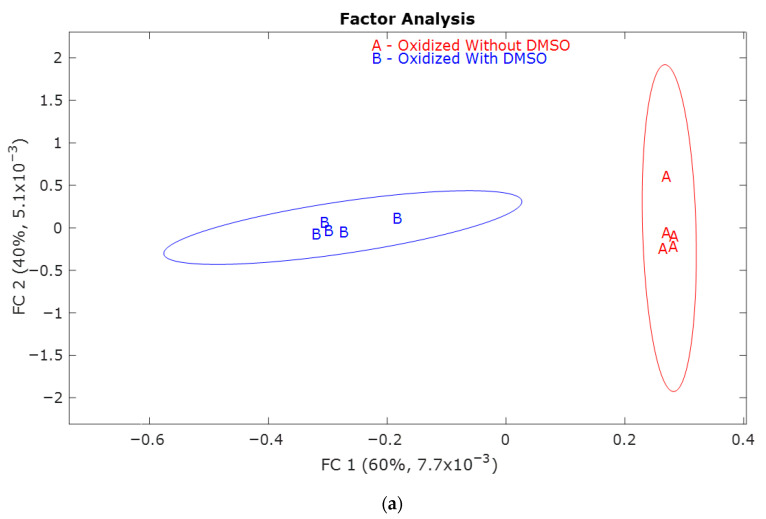
(**a**) Factor analysis to visualize the distributions of the five replicates for $O_{0.6}$ (red, without DMSO) and $O_{0.6}^{D}$ (blue, with DMSO) oxidized samples; (**b**) Retention time loadings, and (**c**) Mass spectral loadings.

**Table 1 molecules-30-04305-t001:** Oxidation Products for Sample $O_{1.0}$ (1.0 V vs. Hg/HgO, no DMSO).

Retention Time (min)	QRMf(%)	Similarity(0–1)	Compound Name	Structure
3.67	1	0.64	5-Hexen-2-one, 5-methyl	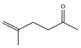
4.26	36	0.90	Acetic acid, ethoxy-, ethyl ester	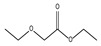
7.20	5	0.64	Cyclohexanecarboxylic acid, -methyl ester	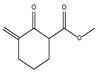
8.01	2	0.93	Propanoic acid, 2-methyl-2, 2-dimethyl-1-(2-hydroxy-1-methylethyl)-, propyl ester	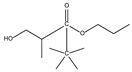
8.14	90	0.96	Propanoic acid, 2-methyl-, 3-hydroxy-2,4,4-trimethylpentyl ester	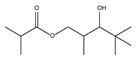
8.97	8	0.68	2-Oxetanone	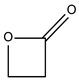
9.01	38	0.76	Phenol, 2,6-bis(1,1-dimethylethyl)-	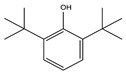
9.55	8	0.98	Furan, 2-butyltetrahydro-	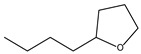
9.88	81	0.91	Dodecanoic acid, 2,3-bis(acetyloxy)propyl ester	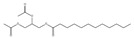
10.78	53	1.00	Isopropyl myristate	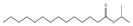
11.06	96	0.89	1,2-Benzenedicarboxylic acid, bis(2-methyl propyl) ester	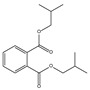
11.57	99	0.83	1,2-Benzenedicarboxylic acid, dibutyl ester	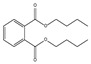

**Table 2 molecules-30-04305-t002:** Oxidation Products for Sample $O_{1.0}^{D}$ (1.0 V vs. Hg/HgO, with DMSO).

Retention Time (min)	QRMf(%)	Similarity(0–1)	Compound Name	Structure
5.29	99	0.90	Pentane 2,3-dimethyl	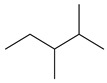
7.24	0.08	1.00	Butanal, 4-hydroxy-3-methyl	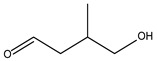
8.01	5	0.94	Propanoic acid, 2-methyl-,2,2-dimethyl-1-(2-hydroxy-1-methylethyl)-, propyl ester	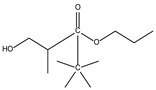
8.14	100	0.96	Butanoic acid, 3-hydroxy-2,2-dimethyl-, ethyl ester	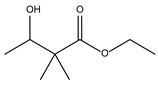
8.38	48	1.00	2,4,7,9-Tetramethyl-5-decyn-4,7-diol	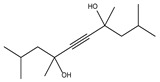
9.01	12	0.57	Phenol, 2,6-bis(1,1-dimethylethyl)-	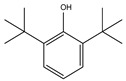
9.55	32	0.98	Furan, 2-butyltetrahydro-	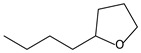
10.78	24	0.86	Isopropyl myristate	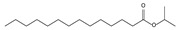
11.06	94	0.88	1,2-Benzenedicarboxylic acid, bis(2-methylpropyl) ester	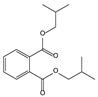
11.54	96	1.00	1,2-Benzenedicarboxylic acid, butyl 2-ethylhexyl ester	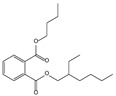

**Table 3 molecules-30-04305-t003:** Oxidation Products for Sample $O_{0.8}$ (0.8 V vs. Hg/HgO, Without DMSO).

Retention Time (min)	QRMf%	Similarity	Compound Name	Structure
8.14	13	0.70	Propanoic acid, 2-methyl-, heptyl ester	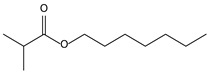
9.54	42	0.83	Furan, 2-butyltetrahydro-	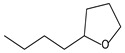
10.76	27	0.87	9-Octadecenoic acid (*Z*)	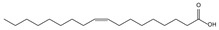
11.53	60	0.96	1,2-Benzenedicarboxylic acid, dibutyl ester	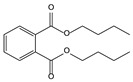

**Table 4 molecules-30-04305-t004:** Oxidation Products for Sample $O_{0.8}^{D}$ (0.8 V vs. Hg/HgO, With DMSO).

Retention Time (min)	QRMf (%)	Similarity (0–1)	Compound Name	Structure
5.38	82	0.86	Cyclohexene, 1-methyl-4-(1-methylethenyl)	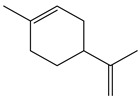
9.01	40	0.84	Phenol, 2,4-bis(1,1-dimethylethyl)-	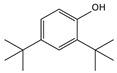
9.54	24	0.99	Furan, 2-butyltetrahydro-	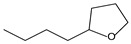
11.06	98	0.88	1,2-Benzenedicarboxylic acid, bis(2-methylpropyl) ester	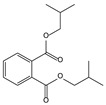

**Table 5 molecules-30-04305-t005:** Oxidation Products for Sample $O_{0.6}$ (0.6 V vs. Hg/HgO, Without DMSO).

Retention Time (min)	QRMf %	Similarity	Compound Name	Structure
5.38	82	0.86	Cyclohexene, 1-methyl-4-(1-methylethenyl)	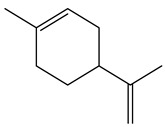
8.02	34	0.90	1,3-Butanediol, 2-methyl-	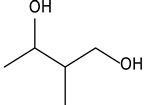
8.14	67	0.96	Butanoic acid, 3-hydroxy-2,2-dimethyl-, hexyl ester	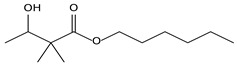
9.01	40	0.84	Phenol, 2,4-bis-(1,1-dimethylethyl)-	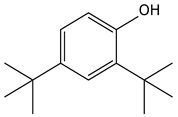
9.54	46	0.84	Furan, 2-butyltetrahydro-	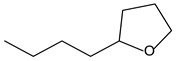

**Table 6 molecules-30-04305-t006:** Oxidation Products for Sample $O_{0.6}^{D}$ (0.6 V vs. Hg/HgO, With DMSO).

Retention Time (min)	QRMf	Similarity	Compound Name	Structure
8.02	10	0.93	Propanoic acid, 2-methyl- 2,2-dimethyl-1-(2-hydroxy-1-methylethyl)-, propyl ester	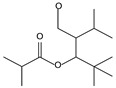
8.09	100	0.95	Butanoic acid, 3-hydroxy-2,2-dimethyl-, hexyl ester	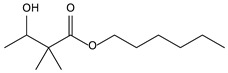
9.01	100	0.58	Phenol, 2,4-bis(1,1-dimethylethyl)-	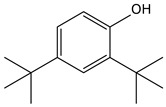
9.54	5	0.95	Furan, 2-butyltetrahydro-	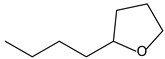

**Table 7 molecules-30-04305-t007:** Details About the Samples, Including Names (Designated Solely for Internal use), Oxidation Potentials, and the Presence or Absence of DMSO. In the table, ‘O’ denotes oxidized, and ‘D’ indicates DMSO presence.

Sample Number	Sample Name	Oxidation Potential (V) vs. Hg/HgO	Presence (*D*)/Absence of DMSO
** 1. **	$O_{1.0}$	1.0	-
** 2. **	$O_{1.0}^{D}$	1.0	D
** 3. **	$O_{0.8}$	0.8	-
** 4. **	$O_{0.8}^{D}$	0.8	D
** 5. **	$O_{0.6}$	0.6	-
** 6. **	$O_{0.6}^{D}$	0.6	D

## Data Availability

The original contributions presented in this study are included in the article/Appendix A. Further inquiries can be directed to the corresponding author.

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
