# Peer review of "Multiway Analysis of the Electrochemical Oxidation Pathway of a Lignin Using Chemometrics"

_molecules, 2025, doi:10.3390/molecules30214305_

Round 1

Reviewer 1 Report

Comments and Suggestions for Authors

This work predicted that both electrocatalyst and •OH radicals contribute to the electrochemical depolymerization of lignin. The quality of this manuscript is not good. Major revision is suggested.

1. The main degradation produts need be described in "Abstract".

2. The CAS number of lignin need be provided.

3. The equipment image of electrochemical conversion of the lignin is required to give. The manufacturer of reaction equipment need be given.

4. This work lacks of enough and in-depth discussion about electrochemical oxidation mechanism. Based on their findings, the scheme about electrochemical oxidation mechanism need be provided.

5. Most compounds fragmented from lignin with DMSO (Table 2) are different from the compounds formed without DMSO (Table 1). Please clarify the reason clearly.

6.  If free hydroxyl radicals can be detected? The direct electrochemical reactions in the electrochemical depolymerization process need be well interpreted.

7. The main products derived from lignin need be quantified. 

8. Other related works need be cited and compared. the novelty of this work need be well highlighted.

Author Response

This work predicted that both electrocatalyst and •OH radicals contribute to the electrochemical depolymerization of lignin. The quality of this manuscript is not good. Major revision is suggested.

  1. The main degradation products need to be described in "Abstract".

Response: Thank you for the suggestion. We have included the main products in the abstract now.

  1. The CAS number of lignin need be provided.

Response: Thank you for the suggestion. The CAS number has been added in the ‘Materials’ section of the manuscript.

  1. The equipment image of electrochemical conversion of the lignin is required to give. The manufacturer of reaction equipment need be given.

Response: We thank the reviewer for this valuable suggestion. In the revised manuscript, we have added the manufacturer of the electrochemical reactor in the Experimental Section. We have also clarified that electrochemical lignin oxidation was performed in a 10 cm² single-cell reactor whose schematic and operational details (flow field structure, electrode configuration, and temperature control) are already reported in Figure 1 of our previous work. (J. Electrochem. Soc., 166, E317–E322, 2019), which is now cited as reference [65]. We believe this sufficiently addresses the request for reactor details while avoiding duplication of a previously published figure.

  1. This work lacks of enough and in-depth discussion about electrochemical oxidation mechanism. Based on their findings, the scheme about electrochemical oxidation mechanism need be provided.

Response: We agree with the reviewer that expanding the mechanistic discussion improves the manuscript. Accordingly, we have added further details on the electrochemical oxidation in the ‘experimental section’. We have also cited our work, which includes details on the equipment for electrochemical oxidation.

NaderiNasrabadi et al. (J. Electrochem. Soc., 166, E317–E322, 2019), cited as reference [65].

  1. Most compounds fragmented from lignin with DMSO (Table 2) are different from the compounds formed without DMSO (Table 1). Please clarify the reason clearly.

Response: Thank you for the suggestion. While we explained the reason why compounds are different with and without DMSO, we have added some lines (157-161) that specifically talk about the reason more clearly.

  1. If free hydroxyl radicals can be detected? The direct electrochemical reactions in the electrochemical depolymerization process need be well interpreted.

Response: We appreciate the reviewer’s insightful comment. In our previous work on the electrochemical oxidation of lignin model compound [64], we discussed the mechanistic possibilities involving both radical-mediated reactions and direct electrode-surface oxidation. While hydroxyl radicals (OH ) are commonly proposed as reactive intermediates in electrochemical oxidation systems, their direct detection is challenging due to their short lifetimes. In our current study, we did not directly measure free OH radicals.

We used DMSO as a radical quencher to quench the hydroxyl radicals formed during the electrochemical oxidation. We compared the samples with (assuming no free OH radicals) and without DMSO (with free OH radicals). We tested these two samples across three different oxidation potentials and observed significant differences between samples oxidized with and without DMSO  (observed in Factor plot in Figure 2(a), 5(a) and 8(a). The differences observed in the fragment pattern in mass spectral loadings further confirm the differences. We also found different oxidized compounds with and without DMSO (Tables 2 and 3). For this reason, we proposed that both OH radicals and direct electrochemical oxidation are involved in the electrochemical depolymerization of lignin.

  1. The main products derived from lignin need be quantified.

Response: We thank the reviewer for this suggestion. In the present work, our primary goal was to interpret the depolymerization mechanism of lignin—specifically, whether the reaction proceeds through direct electrochemical oxidation at the electrode surface or through attack by hydroxyl radicals. While we did not quantify the compound, the identification process was based on both QRMf values and similarity scores in the GC–MS library search, ensuring reliable assignments of the major peaks. Quantification would require calibration with standards for each product, which is beyond the scope of this mechanistic study. Instead, we applied factor analysis to statistically differentiate between lignin samples with and without DMSO.

  1. Other related works need be cited and compared. the novelty of this work need be well highlighted.

Response: We thank the reviewer for this suggestion.  We have rewritten the novelty in the last paragraph of the introduction section. We also added our previous work on ‘Multiway Analysis of the Electrochemical Oxidation Pathway of a Lignin Model Compound, Benzyl Phenyl Ether, with Chemometrics’. Cited as reference [64].  We have also added two other citations recommended by another reviewer as references [6] and [30].

The novelty of the paper has been reemphasized in the last paragraph of  the introduction section.

[64] Sah, G.; Chen, Z.; Staser, J. A.; Harrington, P. de B. Multiway Analysis of the Electrochemical Oxidation Pathway of a Lignin Model Compound, Benzyl Phenyl Ether, with Chemometrics. J. Anal. Test. 2025, 9, 346.

[6]Verdía Barbará, P., et al., Recent Advances in the Use of Ionic Liquids and Deep Eutectic  Solvents for Lignocellulosic Biorefineries and Biobased Chemical and Material Production. Chemical Reviews, 2025.

[30] Luo, J. and T.L. Liu, Electrochemical valorization of lignin: Status, challenges, and prospects. Journal of Bioresources and Bioproducts, 2023. 8(1): p. 1-14.

Reviewer 2 Report

Comments and Suggestions for Authors

1. The authors must specify the exact concentration of DMSO used as the radical scavenger. This critical information is missing from the experimental section and is essential for reproducibility and proper interpretation of the mechanism. Please provide the concentration used and justify this choice based on literature precedent for similar radical scavenging studies.

2. The manuscript lacks essential characterization details for the lignin sample. The authors state it was purchased from Sigma-Aldrich but provide no information about its type (kraft, organosolv, etc.), molecular weight distribution, or purity. This information is crucial for readers to understand the substrate and for reproducibility of the work.

3. Many compound identifications rely on mass spectral library matches with low similarity scores (some as low as 0.52). The authors should acknowledge the limitations of these identifications and consider grouping low-confidence matches by chemical class rather than reporting specific compound names. Additionally, retention indices should be provided where possible to support compound identification.

4. The statistical validation of the chemometric analysis needs strengthening. While confidence ellipses are shown, the authors should include quantitative statistical tests to support claims of significant group separation. Cross-validation results for the factor analysis models would also enhance the robustness of the conclusions.

5. Figure quality requires improvement throughout the manuscript. Figure 1 shows overlaid chromatograms that are difficult to distinguish due to poor color contrast. The mass spectral loadings figures (Figures 3, 6, 9) need numerical scales and better axis labeling for proper interpretation. Additionally, there appears to be a formatting error in line 140 ("Table 2. b)") that should be corrected

6. Authors should also cite https://doi.org/10.1021/acs.chemrev.4c00754 and 

https://doi.org/10.1016/j.jobab.2022.11.003

Author Response

The authors must specify the exact concentration of DMSO used as the radical scavenger. This critical information is missing from the experimental section and is essential for reproducibility and proper interpretation of the mechanism. Please provide the concentration used and justify this choice based on literature precedent for similar radical scavenging studies.

Response: We appreciate the reviewer's feedback. The experimental section has been updated to include DMSO concentration.

This concentration was not chosen based on our previous findings [64], where its efficacy in scavenging radicals was confirmed during the analysis of a lignin model compound. Its proven performance in a directly analogous system provides a strong precedent for its use in this study.

[64] Sah, G.; Chen, Z.; Staser, J. A.; Harrington, P. de B. Multiway Analysis of the Electrochemical Oxidation Pathway of a Lignin Model Compound, Benzyl Phenyl Ether, with Chemometrics. J. Anal. Test. 2025, 9, 346.

  1. The manuscript lacks essential characterization details for the lignin sample. The authors state it was purchased from Sigma-Aldrich but provide no information about its type (kraft, organosolv, etc.), molecular weight distribution, or purity. This information is crucial for readers to understand the substrate and for reproducibility of the work.

Response: We thank the reviewer for highlighting the need for more detailed characterization of the lignin substrate. We have revised the manuscript to include the specifications provided by the supplier in ‘Material’ section.

  1. Many compound identifications rely on mass spectral library matches with low similarity scores (some as low as 0.52). The authors should acknowledge the limitations of these identifications and consider grouping low-confidence matches by chemical class rather than reporting specific compound names. Additionally, retention indices should be provided where possible to support compound identification.

Response: Thank you for your comment. We do not rely solely on the similarity score for our library search; we have also integrated a quantitative reliability metric (QRMf) that statistically assesses the reliability of the match. You can find further details about the similarity score and QRMf in the ‘Instrumental Analysis’ section.

  1. The statistical validation of chemometric analysis needs strengthening. While confidence ellipses are shown, the authors should include quantitative statistical tests to support claims of significant group separation. Cross-validation results for the factor analysis models would also enhance the robustness of the conclusions.

Response: We thank the reviewer for the suggestion. Our confidence in the interpretation of the mechanism has not come only from the score plot. We have shown the mass spectral loadings for each of the score plots and the oxidized products formed with and without DMSO. One can observe different patterns of fragmentation of the oxidized product with and without DMSO in each of the mass spectral loading plots. Furthermore, we have already applied this concept to our recently published work on the lignin model compound, benzyl phenyl ether, where we observed a similar mechanism. We have now cited our recently published work in the current manuscript as reference 64.

[64] Sah, G.; Chen, Z.; Staser, J. A.; Harrington, P. de B. Multiway Analysis of the Electrochemical Oxidation Pathway of a Lignin Model Compound, Benzyl Phenyl Ether, with Chemometrics. J. Anal. Test. 2025, 9, 346.

  1. Figure quality requires improvement throughout the manuscript. Figure 1 shows overlaid chromatograms that are difficult to distinguish due to poor color contrast. The mass spectral loadings figures (Figures 3, 6, 9) need numerical scales and better axis labeling for proper interpretation. Additionally, there appears to be a formatting error in line 140 ("Table 2. b)") that should be corrected

Response:

Figure 1:

Figure 1 has been changed, and the new figure includes the zoomed image of a chromatogram at retention time 10.78. This will help the reader understand that there are five replicates for both oxidized without DMSO (red) and oxidized with DMSO (blue).

Figures 3, 6, 9:

Thank you for your feedback. The Y-axis is labeled "Summed Rotated Variable Loadings" as it represents the aggregated contribution of each m/z value to the rotated principal components after PCA. The Y-axis values are calculated by transforming the original mass spectra using factor analysis. These transformed values are scaled to capture the variance of the data, and the summation reflects the combined contribution of the m/z values across the components. The numerical scale on the Y-axis corresponds to the magnitude of these rotated loadings, which are scaled and centered to provide meaningful visualization. The X-axis is labeled "m/z" to correspond to the mass-to-charge ratio, and it covers the range of 50–300 m/z. The numerical scales are adjusted to ensure proper representation of the data.

The formatting error in line 140 ("Table 2. b)") has been addressed.

  1. Authors should also cite https://doi.org/10.1021/acs.chemrev.4c00754 and 

https://doi.org/10.1016/j.jobab.2022.11.003
Response: Thank you for the suggestion. Both the recommended papers have been cited in the revised manuscript.

Reviewer 3 Report

Comments and Suggestions for Authors

As commented in the references section, the manuscript is not suitable for peer-reviewing. Bibliography should be carefully addressed to provide scientifically sound support to the manuscript.

Author Response

Reviewer 3

As commented in the references section, the manuscript is not suitable for peer-reviewing. Bibliography should be carefully addressed to provide scientifically sound support to the manuscript.

Bibliography has been corrected and properly formatted. Sorry about that in the first draft.

Reviewer 4 Report

Comments and Suggestions for Authors

iThenticate report evidenced a percent match of 26%, try to solve this aspect!

Please check the editing of the text (for example, the period before the brackets).

The quality of Figure 2a, Figure 5a, as well as of Figure 9a needs to be  improved.

The topic is very actual due to the growing interest for valorization of biomass and its components, having in mind the petroleum crisis.
The electrochemical oxidation mechanism is challenging because lignin has an aromatic tridimensional structure, the elucidation of reaction mechanisms being quite difficult.
The authors electrochemically oxidized some lignin compounds using a nickel-cobalt (Ni-Co) electrocatalyst, concluding that both electrocatalyst and •OH radicals contribute to the electrochemical depolymerization of lignin.
The conclusions are consistent with the obtained results.
The references are listed according to the discussion of authors results and sustain them.

Author Response

Reviewer 4

Please check the editing of the text (for example, the period before the brackets).

Response: Thank you for the comment. We have resolved this issue.

The quality of Figure 2a, Figure 5a, as well as of Figure 9a needs to be  improved.

Response: Figures 2a, 5a, and 9a have been presented more clearly. They have been enlarged for easier reading.

The topic is very actual due to the growing interest for valorization of biomass and its components, having in mind the petroleum crisis.
The electrochemical oxidation mechanism is challenging because lignin has an aromatic tridimensional structure, the elucidation of reaction mechanisms being quite difficult.
The authors electrochemically oxidized some lignin compounds using a nickel-cobalt (Ni-Co) electrocatalyst, concluding that both electrocatalyst and •OH radicals contribute to the electrochemical depolymerization of lignin.
The conclusions are consistent with the obtained results.
The references are listed according to the discussion of authors results and sustain them.

Response: Thank you.

Reviewer 5 Report

Comments and Suggestions for Authors

Dear Authors,

The presented work is very interesting as a new solution for the transformation of lignin into valuable chemical compounds. I have no comments here, but please note the text editing, which should be improved. Comments are attached.

Best regards.

Author Response

Response: Thank you for pointing out the formatting details. We have addressed them.

Abstract: Please check for double spaces in the text.

Keywords –> lignin valorization; bimetallic oxidative catalysis; multiway GC-MS analysis; factor analysis; product identification

Response: We have applied the suggestion. Thank you.

Introduction

Line 23: Nickel-Cobalt (Ni-Co) – I suggest lowercase element names

Response: Thank you for suggestion. We will keep as it is.

Line 38: timeframe.[1, 2] Currently – no space and double space

Line 40: damage. One – double space

Line 42: lignin.[3-5] Governments – no space and double space

Line 43: fuels. By – double space

Line 45: 2005.[6, 7] About – no space and double space

Line 46: [8, 9] Because – double space, [8, 9]. Because

Line 48: Because lignin is rich in phenylpropanoid content and has significant annual production from the pulp and paper industry, it has been considered a feedstock for manufacturing petroleum-based chemicals such as phenolic compounds and aromatic hydrocarbons.[9, 10] -> Because lignin is rich in phenylpropanoid content and has significant annual production from the pulp and paper industry. It has been considered a feedstock for manufacturing petroleum-based chemicals such as phenolic compounds and aromatic hydrocarbons [9, 10].

Response: Thank you for the suggestion. We will keep it as it is.

Line 49: [9, 10] Lignin - double space

Line 50: water.[11] However - no space and double space

Line 53: mills.[12] A low-cost - no space and double space

Line 54: phenols. In addition - double space

Line 58: (LMWAs)[9] - no space

Line 58: hydrodeoxygenation[15, 16], pyrolysis[17], microwave-assisted depolymerization[18, 19],

gasification in supercritical water[20, 21], oxidative lignin depolymerization[22, 23], hydrocracking[24,

25], and hydrothermal fragmentation and condensation[26]. – please leave a space between words and

the quote => such as hydrodeoxygenation [15, 16],

Line 63: hydrolysis.[9, 27, 28] The main - no space and double space

Line 65: hydrolysis.[11] Although - no space and double space

Line 68: system.[9, 27] -> system [9, 27].

Line 69: process. Bailey - double space

Line 70: mid-1940s.[29, 30] Several - no space and double space

Line 71: lignin.[31-41] Compared - no space and double space

Line 72: oxidation[9, 32, 42-46] - no space

Line 73: friendly.[47, 48] The participating - no space and double space

Line 74: byproducts.[45, 49] An added - no space and double space

Line 76: cathode.[50, 51] Additionally - no space and double space

Line 77: neutrality.[52] -> neutrality [52].

Line 80: crucial. This - double space

Line 82: lignin.[32] We later -> lignin [32]. We later…[]

Line 82: shell alloy[31] -> shell alloy [31]

Line 83: electrocatalysts[53] -> electrocatalysts [53]

Line 88: analysis.[54-57] For instance -> analysis [54-57]. For instance

Line 88: Prothmann et al.[54] -> Prothmann et al. [54], no spaces

Line 90: UHPLC/HRMSn -> what does n in the index mean?

Line 92: reader.[57] Similarly -> reader [57]. Similarly, remove the double space.

Line 92: Gonçalves[55] -> Gonçalves [55]

Line 95: Fink and colleague[56] -> Fink et al. [56]

Line 97: Wang and colleagues -> Wang et al.

Line 98: electrodes.[58] They -> electrodes [58]. They

Line 98: lignin. They -> double space

Line 99: Quiroz et al.[59] -> Quiroz et al. [59]

Line 101: Cabrera et al.[31] -> Cabrera et al. [31]

Line 102: lignin. Zhu – double space

Line 104: (GC-MS).[35] -> (GC-MS) [35].

Line 105: reactions.[60, 61] -> reactions [60, 61].

Line 106: compounds.[29] -> compounds [29].

Line 107: Numerous studies[31, 34, 58, 59] -> Numerous studies [31, 34, 58, 59]

Line 108: radical attack. However – double space

Line 110: surface. Our - double space

Line 111: lignin. By – double space

Line 114: radicals. Beyond – double space

Abstrakt: proszę sprawdzić czy nie ma podwójnych spacji w tekście.

Please pay more attention to proofreading and editing the text. There are many visible editorial errors.

The citation has a distinctive notation and should be written as follows:

- if it ends a sentence, the correct notation should be: system.[9, 27] -> system [9, 27].

- if the citation appears in the middle of a sentence, the word after it should be: Numerous studies[31,34, 58, 59] -> Numerous studies [31, 34, 58, 59]. Apply to the entire text, as it contains incorrect

notations throughout.

Line 117: 2. Results -> 2. Results and Discussion

Line 120: DMSO – expand the abbreviation

Response: Thank you. The abbreviation has been given.

Line 128: Figure 1 – remove bold text. Apply to other Figures and Tables throughout the text

Response: Thank you for the suggestion. However, we like to keep bold text for figures and tables.

Line 141: Table 2. b) – please correct

Line 174: Propanoic acid -> propanoic acid

Line 260: Was the lignin dried before processing? Or were any other modifications made to it or was it specially prepared? What was its size? Please list all solvents, not just selected ones.

Response: We have revised the manuscript to include the details about the lignin including type 

We really appreciate the time this review spent. All revisions have been made and the manuscript carefully proofread, revised, and formatted.

Round 2

Reviewer 1 Report

Comments and Suggestions for Authors

This revised version can be accepted.

Reviewer 3 Report

Comments and Suggestions for Authors

Authors did not addressed the reference flaws highlighted in the previous revision. Therefore, I suggest a definitive rejection of the manuscript.

Reviewer 5 Report

Comments and Suggestions for Authors

I have no comments.